# Independent domestication and cultivation histories of two West African indigenous fonio millet crops

Thomas Kaczmarek [1,2,3] ✉, Philippe Cubry [3], Louis Champion [3], Sandrine Causse [1,2], Marie Couderc[3], Julie Orjuela[3], Edak A. Uyoh[4], Happiness O. Oselebe[5], Stephen N. Dachi[6], Charlotte O. A. Adje[7], Emmanuel Sekloka[8], Enoch G. Achigan-Dako [7], Abdou R. Ibrahim Bio Yerima[7,9], Sani Idi Saidou[10,11], Yacoubou Bakasso[11,12], Baye M. Diop[13], Mame C. Gueye[13], Richard Y. Agyare [14], Joseph Adjebeng-Danquah [14], Mathieu Gueye [15], Jan J. Wieringa [16], Yves Vigouroux [3,17] ✉, Claire Billot[1,2,17] ✉, Adeline Barnaud [3,17] ✉ & Christian Leclerc [1,2,17] ✉

Crop evolutionary history and domestication processes are key issues for better conservation and effective use of crop genetic diversity. Black and white fonio (*Digitaria iburua* and *D. exilis*, respectively) are two small indigenous grain cereals grown in West Africa. The relationship between these two cultivated crops and wild *Digitaria* species is still unclear. Here, we analyse whole genome sequences of 265 accessions comprising these two cultivated species and their close wild relatives. We show that white and black fonio were the result of two independent domestications without gene flow. We infer a cultivation expansion that began at the outset of the CE era, coinciding with the earliest discovered archaeological fonio remains in Nigeria. Fonio population sizes declined a few centuries ago, probably due to a combination of several factors, including major social and agricultural changes, intensification of the slave trade and the introduction of new, less labour-intensive crops. The key knowledge and genomic resources outlined here will help to promote and conserve these neglected climate-resilient crops and thereby provide an opportunity to tailor agriculture to the changing world.

Plant domestication provides a valuable framework for understanding how diversity evolves in response to environmental and human pressures[1–3]. Cultivating more genetically diverse crops is recognised as a key to the successful transition towards more sustainable agriculture and food systems[4–6], yet genomic crop diversity studies have primarily been focused on major crops. Meanwhile, there has been very little research on the domestication of sub-Saharan crops, particularly indigenous crops cultivated in traditional smallholder farming systems that have great agronomic and nutritional potential[7,8].

The name fonio encompasses two similar indigenous cereal crops cultivated in West Africa, i.e. *Digitaria exilis* (Kippist.) Stapf and *D. iburua* Stapf. The dark brown spikelets of *D. iburua*, which are lighter in *D. exilis*, explain why they are commonly referred to as black and white fonio, respectively[9,10]. White fonio is the most widespread of these two species, with a distribution stretching from Senegal to Nigeria, and possibly as far as Lake Chad. This species is found in a wide range of agroecological niches[11]. Otherwise, black fonio is cultivated almost solely in the Nigerian highlands, and possibly also in the Atakora

A full list of affiliations appears at the end of the paper. ✉e-mail: thomas.kaczma@gmail.com; yves.vigouroux@ird.fr; claire.billot@cirad.fr; adeline.barnaud@ird.fr; christian.leclerc@cirad.fr

mountains of Benin and Togo[12,13]. It has been recently reintroduced in cropping systems in the Atakora region of Benin.

The domestication history of these two species is still unclear, yet it has seldom been a specific focus of research. First, morpho-botanical studies have primarily revealed that the two fonio millets are associated with distinct wild species that prevail in hot tropical areas worldwide[14,15]. For *D. exilis*, the hypothesised closest wild relative species is *D. longiflora* (Retz.) Pers., which very closely resembles fonio although it bears hairy spikelets[14,16–18]. On the other hand, *D. ternata* (A. Rich.) Stapf is reportedly the most likely wild progenitor of black fonio (*D. iburua*)[10,13,19]. Other wild relatives have also been proposed by Henrard[20], who grouped *D. longiflora* with *D. fuscescens*, and fonio millets with another species, i.e. *D. barbinodis*. These affinities with *D. barbinodis* were nevertheless challenged by a further morpho-botanical study[21]. Moreover, a genetic study using random amplified polymorphic DNA (RAPD) on five *Digitaria* species showed that *D. fuscescens* differed the most from fonio millets relative to the originally hypothesised closest wild relatives[10]. In the absence of other robust phylogenies, it can thus be assumed with confidence that *D. longiflora* and *D. ternata* are the closest wild relatives of fonio millets. However, no genomic studies have focused in detail on this complex of species to understand how close the fonio crops and their wild relatives are to each other.

Flow cytometric analysis of the nuclear DNA indicated that the four species had a similar genome size, all seemed to be tetraploids[22]. Genomic assemblies of *D. exilis* revealed fonio to be a recent (~3 million years) allotetraploid[23,24]. The distinctions between the two pairs of cultivated and wild relative species were highlighted in a few genetic studies that jointly analysed wild and cultivated *Digitaria* grasses with dominant markers such as amplified fragment length polymorphism (AFLP) markers[10,25]. However, these studies did not account for the full range of fonio genetic diversity, as only a couple of accessions were analysed to infer their genetic relatedness. Molecular characterisations with EST-SSR markers and whole genome sequencing were recently carried out on a larger sample of fonio and wild relatives[23,26], but *D. ternata* and *D. iburua*, or both, were not included[23,26]. The aim of the present study was to determine if *D. iburua* and *D. exilis* were derived from one or two domestication events, with or without gene flow, and to reveal the population dynamics that prevailed during this process.

In this work, we assemble a large collection of genomic resources comprising 265 accessions of the four species. It contains whole-genome sequences of *D. iburua* and its wild relative, and a large sampling of *D. exilis* sequences, thereby providing a comprehensive picture of its geographical distribution. We use different complementary methods and genomic approaches to unravel the domestication history of these fonio millets, which is key to tapping the potential of these native species while preserving the diversity necessary for future adaptations. Our results confirm that the two fonio species were domesticated independently, while providing key information that will be essential for further research on these neglected climate-resilient crops.

## Results

### Genomic diversity and population structure revealed two distinct groups

Here we present one of the most geographically complete datasets of cultivated fonio cereals and their associated wild relatives, including 265 genome assemblies (Fig. 1a, Supplementary Table 1, and Supplementary Data 1). We resequenced the genomes of 26 *D. iburua* accessions and of 22 accessions of its closest wild relative *D. ternata*. Moreover, we extended the *D. exilis* sampling to key regions situated at the edge of its geographical distribution range.

The average mapping rates of raw reads on the *D. exilis* reference genome[23] were 99% and 93% for *D. exilis* and *D. longiflora*, respectively (Supplementary Data 2). For *D. iburua* and *D. ternata*, the mapped read percentages were 74% to 68%, respectively (Supplementary Data 2).

The number of unfiltered SNPs was 15,588,339, with a missing rate of 0.64 and 0.03 for *D. iburua* and *D. exilis*, respectively (Supplementary Data 2). After filtering loci with a missing rate > 0.05 and individuals with a missing rate > 0.40, we kept 1,910,119 high-quality bi-allelic SNPs and 247 individuals, with an average missing rate of 0.006 for *D. exilis* ($n = 200$) and 0.09 for *D. iburua* ($n = 21$) (Supplementary Data 3). For the wild relatives, we obtained an average missing rate of 0.06 for *D. longiflora* ($n = 14$) and 0.18 for *D. ternata* ($n = 11$) (Supplementary Data 3). More than a half (52%) of these SNPs were rare variants with a minor allele frequency <0.01 (Supplementary Fig. 1). SNPs were distributed evenly across the 18 chromosomes of the reference assembly, but chromosomes 8A and 8B had a lower SNP density than the other chromosomes (Supplementary Fig. 2).

The nucleotide diversity (π) of cultivated *D. exilis* and *D. iburua* was lower ($P < 2.2 \times 10-16$; two-tailed Welch *t*-test, Supplementary Table 2) than that of their wild relatives *D. longiflora* and *D. ternata* (Fig. 2a, Supplementary Table 3). Moreover, black fonio displayed higher ($P < 2.2 \times 10–16$; two-tailed Welch *t*-test, Supplementary Table 2) genetic diversity than white fonio and a lower reduction in nucleotide diversity compared to its wild relative *D. ternata* (Fig. 2a, Supplementary Table 3). These patterns were also obtained with the Jaccard dissimilarity index that can be viewed as a genome reference-free measure of the population genetic diversity computed with k-mers (Fig. 2b). However, the mean values between *D. iburua* and *D. ternata* were not significantly different ($P = 0.15$; two-tailed Welch *t*-test, Supplementary Table 2). The Watterson's Θ values were higher than π for both species of the *D. exilis/D. longiflora* pair, while being lower for the *D. iburua/D. ternata* pair. A negative Tajima's D value was obtained for *D. exilis/D. longiflora*, while being positive for *D. iburua/D. ternata* (Supplementary Table 3). We identified linkage disequilibrium (LD) decays according to the kb distance for *D. exilis* and *D. longiflora*, while LD very quickly decayed to its minimum value for *D. iburua* and *D. ternata* (Supplementary Fig. 3).

Using SNPs, the first PCA axis (68% of the total variance) differentiated the two cultivated/wild species pairs: *D. exilis/D. longiflora* versus *D. iburua/D. ternata* (Fig. 3a). The second axis (2.6% of the total variance) differentiated the group of cultivated *D. exilis* individuals from their wild relative *D. longiflora*, with East African individuals being more distant than those from West Africa (Fig. 3a, Supplementary Fig. 4). The third axis (1.65% of the total variance) differentiated the cultivated accessions from their associated wild relatives, for both pairs (Supplementary Fig. 4). We noticed that one individual from Benin that was classified as *D. exilis* grouped with *D. iburua*. On the other hand, some accessions identified as *D. iburua* in Nigeria grouped with *D. exilis*, suggesting potential misidentification during the sampling in the region where the two species are reported to co-occur. We corrected this classification in our subsequent analyses. The results obtained with the SNP datasets that had been filtered at different missing data thresholds (5%, 10% and 20%) were consistent (Supplementary Fig. 5).

We also performed a PCA of the four *Digitaria* species using the mapping free approach (k-mer dataset). The results were similar to those obtained in the SNP analysis (Fig. 3b and Supplementary Fig. 6). The two cultivated fonio millet and wild relative pairs were differentiated on the first axis (56% of the total variance). The differentiation between cultivated and wild accessions was associated with the second axis (2.6% of the total variance).

The genetic variation observed among the four species was confirmed by genetic structure analyses conducted with sNMF. The cross-validation error decreased drastically from $K = 2$ to $K = 4$, reaching a minimum at $K = 6$ before sharply increasing at $K = 8$ (Supplementary Fig. 7). The two cultivated/wild species pairs were well differentiated at $K = 2$ (Fig. 3c). At $K = 3$, a *D. longiflora* wild relative group from East Africa was split (Supplementary Fig. 8), while a wild relative group of *D. ternata* individuals from Côte d'Ivoire formed a distinct cluster at $K = 4$

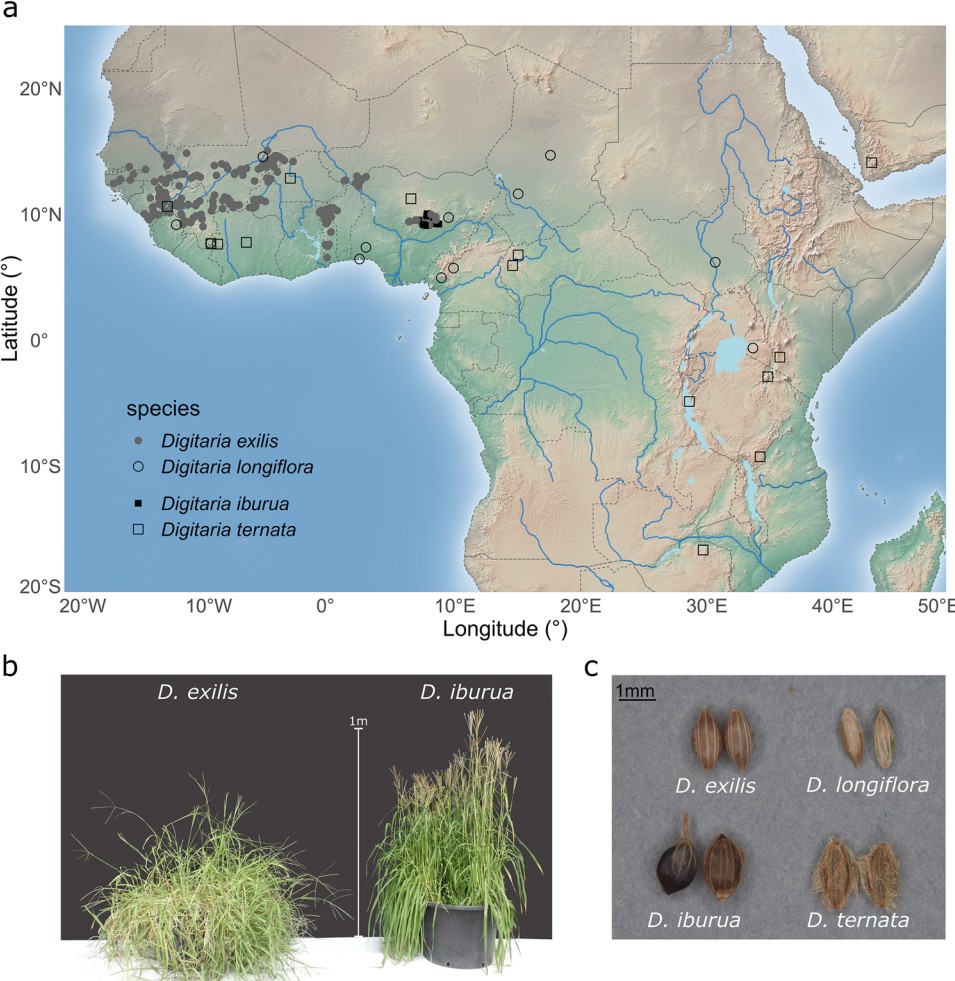

**Fig. 1 | Geographical distribution of *Digitaria* species and phenotypical comparison of cultivated fonio species *D. exilis* and *D. iburua*, and their wild relatives *D. longiflora* and *D. ternata*. a** Map of Africa with the 262 georeferenced accessions of cultivated white fonio (*n* = 203) and black fonio (*n* = 26) with their respective closest wild relative species *D. longiflora* (*n* = 14) and *D. ternata* (*n* = 19). The *D. ternata* species includes three more accessions not represented on the map due to lack of available geographical coordinates (Ethiopia, Kenya, Côte d'Ivoire). **b** Picture of the aerial parts of white fonio (left) and black fonio (right) plants © Sandrine Causse, CIRAD. **c** Picture of fonio grains and those of their associated wild relatives © Sophie Nourissier-Mountou, CIRAD.

(blue colour, Fig. 3c, Supplementary Fig. 8). Two *D. exilis* geographical groups were highlighted at *K* = 5 (Supplementary Fig. 8) and a West African *D. longiflora* group appeared at *K* = 6 (Fig. 3c and Supplementary Fig. 8). Differentiation between *D. ternata* and *D. iburua* was observed from *K* = 7 (Supplementary Fig. 8). At *K* = 8, a third *D. exilis* group, mainly from Nigeria, differed (Supplementary Fig. 8). The structure obtained with the k-mer approach corroborated these results (Supplementary Note 1, Supplementary Figs. 9, 10, and 11).

The pairwise population distance ($D_{xy}$) and the net pairwise distance ($D_a$) between genetic clusters indicated a lower distance between the *D. exilis*/*D. longiflora* pair compared to *D. iburua*/*D. ternata* (Supplementary Table 4). Moreover, the East African *D. longiflora* population and the Côte d'Ivoire *D. ternata* population were more differentiated than the other populations of the same species (Supplementary Table 4).

**Genomic evidence of independent domestication events**

We used three complementary approaches to examine the evolutionary relationships of fonio species and their wild relatives: (i) a genome-wide approach using the Treemix maximum likelihood method[27], (ii) a sliding window computation of neighbour-joining subtrees (Twisst)[28] and (iii) a model-based inference framework based on the site frequency spectrum (SFS) implemented in fastsimcoal v.2.8[29–31].

The inference of the relationship between the four species (four populations) using Treemix[27] distinguished the two cultivated/wild species pairs *D. exilis*/*D. longiflora* and *D. iburua*/*D. ternata*, with substantial genetic drift (>0.2, Fig. 4a). The inference without migration event explained >99.9% of the variance in relatedness between populations. Adding one migration event led to a higher log-likelihood but did not increase the explained variance (Supplementary Fig. 12), suggesting a poorer fit of the model.

Topology weights (Twisst) considering the four species as populations corroborated the distinctiveness of the two wild/cultivated species pairs (Fig. 4b). Indeed, with four populations as input, almost all (99.9%) genomic windows supported the topology distinguishing the two respective fonio millet/wild relative pairs (Fig. 4b).

For both methods, the inferred topologies were robust when more populations were defined based on the genetic structure inferred with sNMF (Supplementary Figs. 13, and 14). These results suggested that white fonio (*D. exilis*) and black fonio (*D. iburua*) millet species had independent histories and domestication patterns, with no evidence of gene flow.

The best tree topology inferred with fastsimcoal showed divergence of cultivated fonio millets from their respective wild relatives (m01 model, Fig. 4c and Supplementary Fig. 15). The addition of a population bottleneck for cultivated species after their divergence

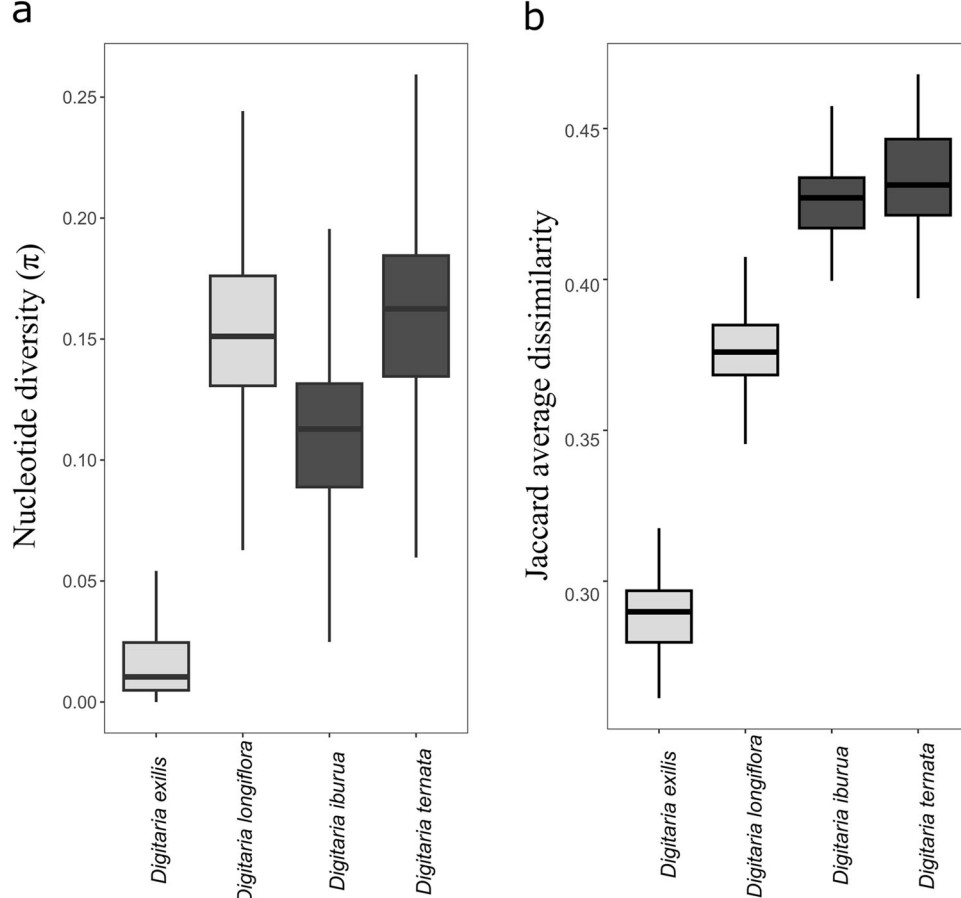

**Fig. 2 | Patterns of genetic diversity in fonio millets and wild relatives, with and without consideration of the *D. exilis* reference genome for calculations.**
**a** Boxplot distribution of nucleotide diversity (π) computed using sliding windows of size 50 kb and step sizes of 10 kb within *D. exilis* (*n* = 199), *D. longiflora* (*n* = 14), *D. iburua* (*n* = 21) and *D. ternata* (*n* = 11). **b** Boxplot distribution of the Jaccard dissimilarity index generated from the 1,000,000 k-mers presence/absence table and calculated within *D. exilis* (*n* = 199), *D. longiflora* (*n* = 13), *D. iburua* (*n* = 21) and *D. ternata* (*n* = 11). The centre line represents the median; the box limits represent the upper and lower quartiles; the upper and lower lines represent 1.5 times the interquartile range. Statistical comparisons between each pair of population are presented in Supplementary Table 2. Source data are provided as a Source Data file.

from their respective wild relatives better supported our data (Supplementary Fig. 16). Parameter estimates showed an old divergence of the wild relatives *D. longiflora* and *D. ternata*, and an older divergence of white fonio from its wild relative compared to black fonio (Fig. 4c and Supplementary Table 5).

## Past population size history and relative timing of domestication

We used the sequentially Markovian coalescent based method smc++[32] to analyse demographic changes associated with domestication, expansion and or contraction of the two cultivated fonio millet species. The effective population size of *D. exilis* steadily declined from more than 20,000 years ago and reached a minimum ~2000 years ago (Fig. 4d). Then the effective population size markedly increased, and then declined again from 500 to 200 years ago (Fig. 4d). Similar trends were noted when the two *D. exilis* populations were differentiated based on genetic clusters, but the onset of the expansion of the Guinean cluster was later than that of the Nigerian cluster (Supplementary Fig. 17). Moreover, only the Nigerian cluster experienced a decrease in population size at 500 years, whereas the effective population size of the Guinean cluster stopped increasing and remained stable (Supplementary Fig. 17). The patterns of the two clusters of the *D. longiflora* wild populations differed. From 60,000 to 30,000 years ago, the East African cluster experienced a marked decrease in population size, while the contrary was

observed for the West African cluster (Supplementary Fig. 17). Then, from 30,000 to 1000 years, the effective population sizes of these populations increased or decreased, respectively, before levelling off (Supplementary Fig. 17). Concerning *D. iburua*, the population decreased sharply from around 20,000 years ago and reached a minimum before *D. exilis*, at around 3000 BP. A lesser increase in the effective population size occurred from this period to 300 years BP. The closest *D. ternata* wild population decreased the same way, but did not increase thereafter like the cultivated population (Supplementary Fig. 17). The effective population size of *D. ternata* from Côte d'Ivoire showed the same decrease until 8000 years, and diverged at 4000 years, with a steep increase until 1000 years ago (Supplementary Fig. 17). However, those results for the *D. iburua* and *D. ternata* populations should be considered with caution as there was high variance between the different runs (Fig. 4d and Supplementary Fig. 17).

## Discussion

Our study highlighted—with different approaches—that cultivated fonio species and their close wild relatives were clearly separated into two distinct pairs. We established the independence of the domestication histories of the two cultivated species, with no gene flow between them. Both species have been cultivated for thousands of years, and they spread from the beginning of the CE era. We discuss how the history of West Africa and sociocultural factors, as well as

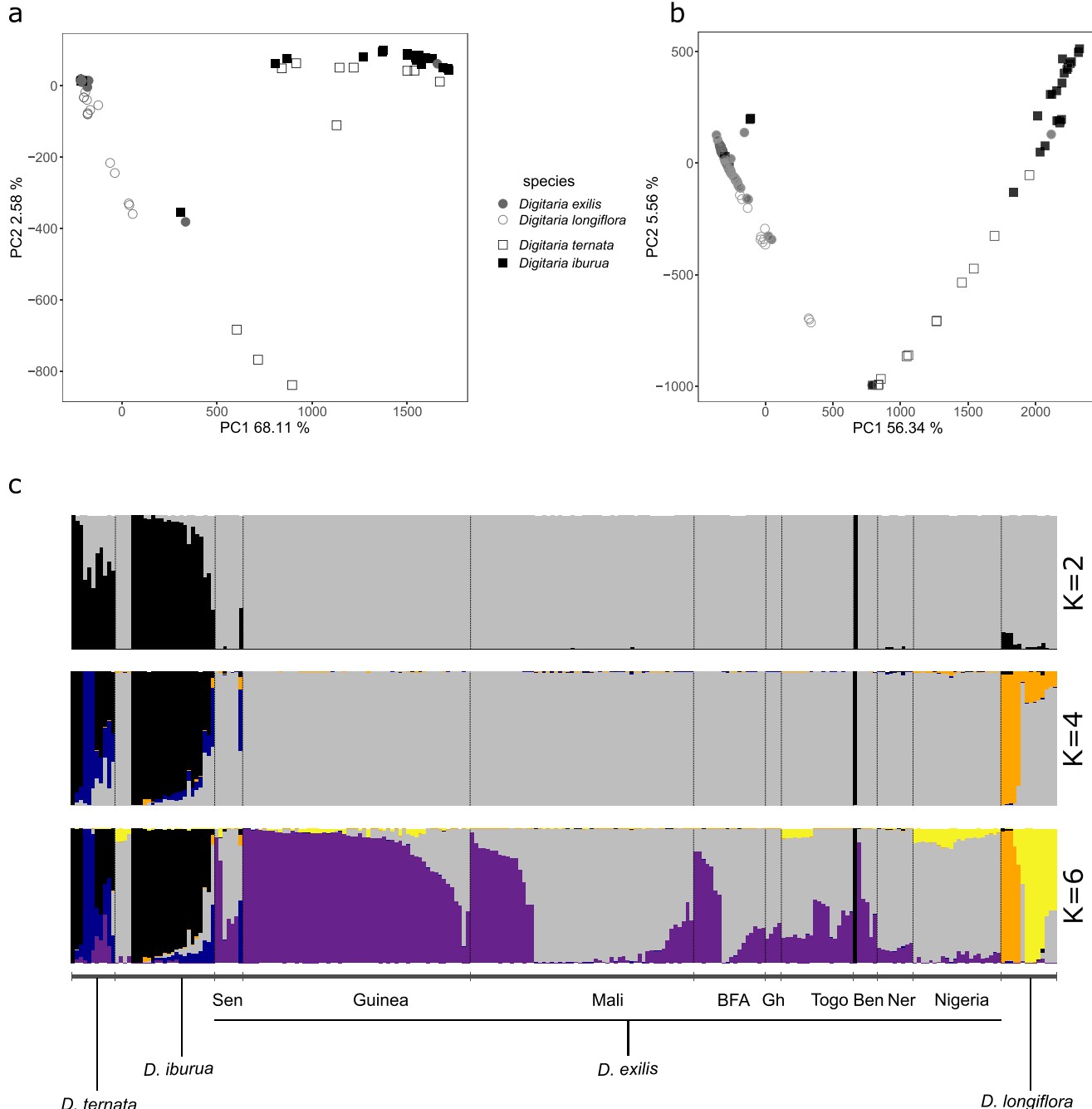

**Fig. 3 | Genetic diversity and structure observed among cultivated fonio millets and their wild relatives (*n* = 247 individuals). a** Principal component analysis focused on the SNP matrix (438,883 SNPs with a missing rate < 0.05) with first and second axes displayed. **b** Principal component analysis focused on the k-mer matrix (1,000,000 k-mers) with first and second axes displayed. **c** Population structure (*K* = 2, *K* = 4 and *K* = 6) of cultivated fonio (*D. exilis* and *D. iburua*) and wild relative (*D. longiflora*, *D. ternata*) accessions estimated with sNMF and the SNP matrix. Each individual is represented by a vertical bar, partitioned into *K* segments representing the proportion of genetic ancestry from the *K* clusters. We ordered individuals according to species and country information in the passport data.

climate change, have impacted the evolution of fonio agrobiodiversity, particularly in the light of the abandonment of their cultivation.

Our joint analyses of the four *Digitaria* species showed that white and black fonio are highly genetically differentiated. The two wild relative species used in this study also clustered with their respective cultivated species, and thereby two distinct pairs were identified. The findings obtained via the k-mer approach, which directly relies on the sequencing reads, were particularly relevant as *D. iburua* and *D. ternata* did not map well on the *D. exilis* reference genome. This mapping-free approach served as a validation criterion for using the SNP dataset

produced with the available reference genome[23]. Congruent results were obtained with both approaches, regardless of the use of PCA analyses and genetic clustering methods. The genetic diversity of black fonio (*D. iburua*) and its wild relative (*D. ternata*) was higher than for white fonio (*D. exilis*) and *D. longiflora*. This result was even more pronounced with the Jaccard index computed with k-mers. The use of the *D. exilis* reference genome, which differs markedly from the *D. iburua* and *D. ternata* genomes, probably underestimated their genetic diversity, since low mapping rates may lead to the loss of genetic variation detected during SNP calling. These differences in genetic

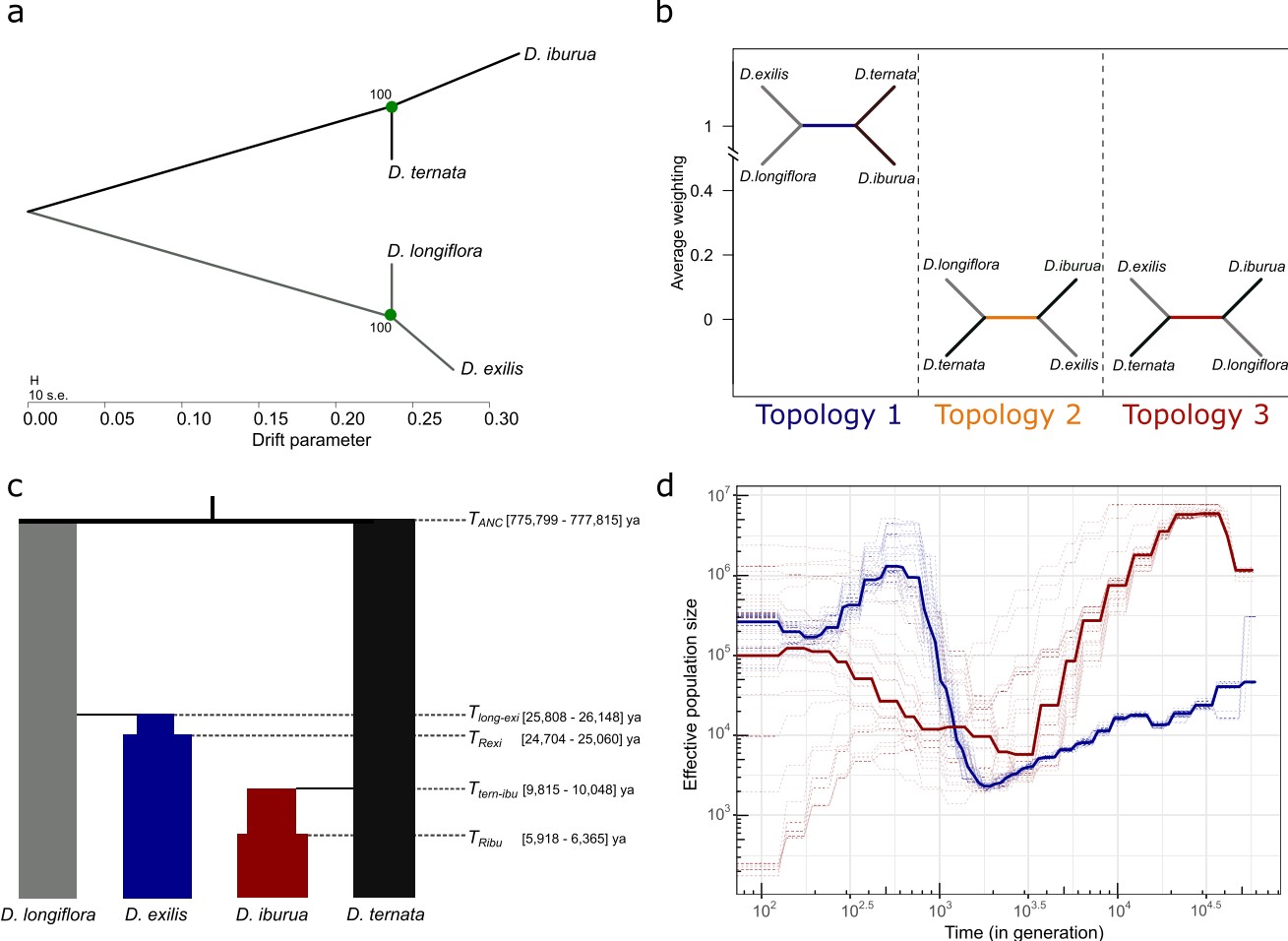

**Fig. 4 | Inference of independent domestication events of cultivated white and black fonio, with divergence without gene flow, and reconstruction of past demographic histories. a** Best Treemix model obtained with the four populations corresponding to the species identifications in the passport data, corrected for potential misidentifications noted in the PCA analyses. Bootstrapped values are shown for the two nodes. **b** Topology weighting analysis of sliding window neighbour-joining trees computed with four populations, using the West African genetic cluster of *D. longiflora* (*n* = 8), the *D. ternata* group (*n* = 6), the *D. iburua* genetic group (*n* = 19) and a subsample of *n* = 21 *D. exilis* individuals representative of the species diversity and geographical distribution. **c** Most likely scenario of

divergence of white and black fonio and *Digitaria* wild relatives inferred with fastsimcoal v.2.8. This scenario assumed an increase in population size after the population bottleneck following divergence. Divergence time estimations are shown, with their 95% confidence interval in brackets. **d** Inference of the effective population size history of cultivated white and black fonio with smc++. The same *D. exilis* (*n* = 21) and *D. iburua* (*n* = 19) individuals as in b and c were used. Thin lines represent the 30 independent runs of smc estimates performed for the two species, and thick lines represent the median of the estimated sizes for *D. exilis* (blue) and *D. iburua* (red).

diversity could reflect differences in reproduction systems among the two species pairs. In fact, the self-fertilisation rate of *D. exilis* is around 99%[33]. Outbreeding might actually be more pronounced for black fonio and its wild relative, thus increasing their genetic diversity. More severe bottlenecks associated with the domestication and diffusion history of white fonio could also explain its lower diversity compared to black fonio. Given our results, we recommend using k-mers as a validation criterion in further genomic studies on indigenous crops (which often lack a reference genome), but also as a means to study their genetic diversity. Such approaches are receiving increasing attention in the population genomics field when no reference genome is available[34].

We highlighted the importance of genetically characterising white and black fonio. Indeed, we amended the species identifications of some individuals that had been incorrectly documented in the databases. Notably, we found one *D. iburua* accession in Benin, even though recent ethnobotanical survey findings suggested that the species had disappeared. These misidentifications could occur

because fonio species are sometimes not viewed as distinct species or varieties in regions where they are jointly grown[35].

Our results highlighted the genetic uniqueness of each fonio millet, which has two distinct gene pools. This information is crucial for conservation and crop improvement programmes as each species requires tailored strategies to ensure its effective use in agrosystems. It also presents an opportunity for breeding programmes, as key traits such as plant architecture differ between white and black fonio, potentially offering key avenues for crop improvement.

We incorporated several new individuals and thus corroborated the phylogenies obtained in previous studies on only a few accessions using AFLP or RAPD markers[10,18]. The k-mer approach enhanced the distinction of the *D. iburua*/*D. ternata* species pair, and revealed a two group separation within the *D. iburua* species. A reference genome would thus be needed for further studies focused specifically on black fonio genomics. Moreover, the currently published *Digitaria* phylogeny includes a total of 63 species, but white and black fonio are not included[36]. Further studies are thus needed to have a comprehensive

picture so as to be able to accurately assess the relationships within the *Digitaria* lineage.

The domestication of plants and animals can have single or multiple origins[2,37,38]. Apricots (*Prunus armeniaca*), for instance, were domesticated from two distinct wild populations that exchanged genes[2]. On the other hand, *Oryza glaberrima* and *Oryza sativa*, or so-called African and Asian 'rice', respectively, resulted from independent domestication events on two different continents[39]. The fonio millet case is all the more remarkable because, unlike rice, the domestication of black and white fonio millets occurred in the same region of Africa, with no gene flow between them. Based on previous phylogenies and morphological characteristics[10,16–18,23,26], we hypothesised that *D. longiflora* and *D. ternata* were the closest wild relative species. As no fonio millet populations are known to grow outside agroecological systems, it is likely that domestication of those species followed divergence from wild relatives. Simulations and parameter estimations of the most likely evolutionary scenario have indicated an earlier divergence time of *D. exilis* from *D. longiflora* (~25,000 generations ago) than that of *D. iburua* from *D. ternata* (~10,000 generations ago). However, as we probably did not sample the true wild populations from which fonio millets were domesticated, those estimates do not necessarily correspond to the onset of domestication. The pre-defined mutation and recombination rates likely also biased these estimates.

The fact that two independent events led to the emergence of two species that share many phenotypical characteristics and that are currently often found growing in the same areas is remarkable. A possible explanation is that both pre-domesticated wild populations had interesting characteristics for potential consumption, but they were growing in the vicinity of different human communities located in different areas. It has been hypothesised that black fonio (*D. iburua*) might originate from the Aïr mountains in Niger, and moved south to Nigeria with the southward migration of Hausa people due to Saharan desertification[13]. This assumption is in line with the current presence of black fonio only in mountainous areas. North to South migration of people due to the end of the African Humid period has also been reported for pearl millet, domesticated in nowadays northwestern Mali[40]. Concerning white fonio, the more diverse range of environments this species currently occupies suggests that it had a distinct geographical origin or a more complex evolutionary history. Some authors have suggested that its domestication occurred in the Guinea/Mali region, in the vicinity of the Niger River, followed by a secondary diversification during its eastward expansion[17,35]. The substantial genomic resources produced in this study should shed light on the geographical origin(s) of white fonio domestication.

The demographic analyses indicated a decrease in population size that started more than 20,000 generations ago for the two fonio millet populations, then reaching a minimum around 2000 years ago for white fonio and 3000 years ago for black fonio. Around the beginning of the CE era, the effective size of fonio species increased. This population bottleneck pattern, followed by population expansion associated with cultivation, was similar to that observed for other sub-Saharan crops[41,42]. Indeed, the decline in effective population size may not only reflect human harvesting pressure on the pre-domesticated wild grasses, but also environmental degradation due to changing climatic conditions, as we also noted a decline in the effective population sizes of the wild relatives[41,43]. The increased aridity in the 1st millennium BCE favoured the settlement of farming communities around water sources, such as Lake Chad, and Niger and Senegal Rivers[44]. Subsequent population movements may have facilitated the expansion of fonio cultivation throughout West Africa.

Otherwise, we observed high variation between the different black fonio effective population size estimates. This uncertainty was due to the absence of data regarding black fonio regions not covered in our reference genome, which resulted in less precise inferences. A black fonio reference genome will thus be required to enhance the estimates.

Our demographic inferences also revealed that fonio expansion first occurred in the Nigerian cluster. The earliest evidence of fonio derives from the Janruwa C site in Nigeria, where many fonio remains were discovered[45]. Fonio was found there in large quantities (3979 grains, 83% frequency, and a density of 33 grains/L, one of the highest rates reported in West Africa). The arrival and dominance of fonio from early AD coincided with the emergence of new human populations and archaeological artefacts, reflecting a complete cultural shift from the previous period[45]. The origin of these populations remains unknown, but the dominance of fonio in local peoples' diet suggests that its domestication occurred earlier elsewhere. Other sites where fonio evidence was found west of Nigeria were from later periods (end of the first millennium AD and second millennium AD), where fonio appeared only during the latest phases of occupation as a secondary introduction. This coincided with the later expansion we inferred for the Guinean cluster, around AD 1000. Overall, these results suggest an initial fonio domestication in Nigeria during the mid-first millennium AD (250–450 AD), followed by a slow westward diffusion.

These results enhance our knowledge of the origins of agriculture in sub-Saharan Africa and support the view that the domestication of African crops took place in a much more restricted area than postulated by Harlan[46]. The domestication non-centre hypothesis had already been ruled out by the findings of genomic studies on pearl millet[40], African rice[41], and yam[42], i.e. three crops domesticated at three different locations near the Niger River, thereby constituting a West African cradle of domestication. Here we may add a fourth location, around Nigeria, from which fonio may have originated. Our study, therefore, paves the way for gaining greater insight into the spatiotemporal nature of crop domestication in Africa and the shared history between humans and crops. For fonio in particular, the scarcity of archaeobotanical remains in West Africa and the difficulty in differentiating white and black fonio seeds hamper efforts to accurately trace the spatiotemporal evolution of fonio millet agrobiodiversity. Black fonio has not yet been clearly documented in archaeobotanical records. Ancient DNA and morphometric analyses would certainly help unravel the cultivation histories of white and black fonio millets.

Variations in effective population size should be interpreted with caution, as the population structure can lead to incorrect inferences[47]. The inferred times for bottlenecks or population expansion also depend on the mutation rate, which has not yet been determined for fonio. In addition, the power to estimate effective population sizes decreases as the time becomes more recent, but coalescent methods using several haplotypes (from a few dozens to hundreds) provide higher resolution than other methods for inferring changes in population size in the recent past[32]. *D. exilis* experienced a decline in effective population size around 500 years ago, which was also the pattern we noted for *D. iburua* in some smc++ runs. This decline has also been observed in similar studies for African yam[42] (*Dioscorea rotundata*) and African rice (*Oryza glaberrima*)[41]. With fonio, there are now three different signs of a recent loss of indigenous African crops, which could be associated with major social and agricultural changes, intensification of the slave trade, and the introduction of new, less labour-intensive crops such as maize, cassava, and Asian rice. Notably, the effective population size of black fonio was found to have collapsed in some of our smc++ runs. In the Atakora mountains of Togo and Benin, surveys and ethnobotanical studies mainly reported the presence of *D. exilis*, with little or no mention of *D. iburua*[9]. It is possible that white fonio replaced black fonio for social or agronomic reasons[16,48]. For instance, women are highly involved in post-harvesting processes and thus certainly have an influence on the maintenance of fonio farmers' varieties, as revealed in their claims that black fonio grains are harder to husk than white fonio grains[16]. Moreover, Bassari people in Senegal abandoned white fonio cultivation because of social and political choices that were incompatible with current climate change issues[49]. Conversely, early fonio varieties are now tending to

disappear in Guinea due to the fact that precipitation is becoming heavier during the harvesting period[50]. Those varieties could, however, be adapted elsewhere, and therefore, future research should—for conservation and breeding purposes—focus on the adaptive potential of fonio species to climate change. The key knowledge and resources outlined here should enhance future research on these climate-resilient crops and thereby provide an opportunity to tailor agriculture to the changing world.

## Methods

### Plant material
We assembled a large collection of fonio genomic resources, with 265 georeferenced accessions, including 94 new sequences (Fig. 1a, Supplementary Table 1, and Supplementary Data 1). It contains whole-genome sequences of *D. iburua* and its wild relative *D. ternata*, and new *D. exilis* sequences, so now its geographical distribution is likely fully represented. For *D. exilis*, a dataset (Abrouk et al.[23]) of 157 accessions is available in the European Nucleotide Archive (ENA), but some countries are missing or poorly represented. We sampled 46 new *D. exilis* accessions originating from Senegal, Benin, Nigeria and Niger to enhance the overall distribution coverage of the species. Our new *D. exilis* collection now consists of a total of 203 accessions from nine countries. For *D. iburua*, we sequenced 26 new samples collected in Nigeria. For the wild relatives, we used the 14 *D. longiflora* accessions analysed in Abrouk et al.[23] and collected 22 accessions from *D. ternata* specimens stored at the National Museum of Natural History in Paris (France) and in CIRAD herbariums[51].

### Library preparation and whole genome sequencing
Total genomic DNA extractions were performed from fresh leaves by an automated method adapted from Risterucci et al.[52] on Biomek FXP (Beckman, Coulter, CA, USA) and using the NucleoMag Plant Kit (Macherey-Nagel, Germany). For *D. ternata* herbarium specimens, DNA was extracted using the modified protocol of Doyle and Doyle[53] (Supplementary Method 1). We constructed paired-end DNA libraries using a homemade library construction approach with a double indexing protocol corresponding to a Tag sequence of six nucleotides on the 5′ DNA end and an index sequence of eight nucleotides on the 3′ DNA end (Illumina Unique Dual Index). This protocol was used to avoid contaminated or recombined reads that would not be correctly assigned during the demultiplexing step. The libraries of the 94 new genomes were sequenced using an Illumina NovaSeq 6000 system with a targeted 10X sequencing depth. A duplicate accession was added as positive sequencing control.

### Raw reads and alignment processing
For the 94 new sequences, the read quality was controlled at each step of the read processing pipeline with FastQC v.0.11.9 software[54]. We checked the purity and quality of the fastq read assignations with a home-made *demultadapt.py* script. This script eliminates forward reads having the wrong Tag sequence assigned to each individual during library preparation. The corresponding reverse reads were then re-associated with the forward reads with a home-made *repairing.sh* script. We removed sequencing adapters, low quality sequences and reads of <35 bp length with Cutadapt v. 3.1[55] using the following parameters: -O 5 -q 20 -m 35 -nextseq-trim 20. The -nextseq-trim parameter was used to correct optical errors due to Novaseq sequencing that could increase the proportion of G bases at the end of a sequencing cycle. The cleaned fastq reads of the 171 accessions from Abrouk et al.[23] were downloaded from ENA with the *enaGroupGet.py* script.

We used bwa-mem2 v.2.2.1[56] for mapping of raw reads on the *D. exilis* reference genome[23]. The alignment was done with the -R and -M options to keep the read group IDs in the output files (*.sam*) and to mark supplementary reads as secondary. Output SAM files were sorted with the picard SortSam algorithm, which produced coordinate-sorted

BAM files. We only kept and indexed properly paired reads using samtools v.1.14[57] with the *view* and *index* algorithms. Alignment summaries were obtained with samtools flagstats and samtools depth. Accessions with a mean depth <1 ($n = 16$) were removed and not considered in the subsequent bioinformatics process.

### SNP and genotype calling, filtering of SNPs and individuals
We used GATK v.4.2.6.1[58] to perform SNP and genotype calling. Duplicate reads were flagged with MarkDuplicates before calling the haplotypes by chromosome with the HaplotypeCaller run in GVCF mode. For each of the 18 chromosomes, we created a genomic database of the accessions with GenomicsDBImport. These databases were used as input for the joint-genotyping step with GenotypeGVCFs.

We removed indels and kept biallelic SNPs with gatk SelectVariants and VCFtools v.0.1.16[59], respectively. We applied hard filters to the biallelic SNPs according to GATK guidelines by filtering SNPs with low (<500) and high (>14,000) depth summed across all samples. Clusters of three SNPs within 10 bp were removed and we also excluded SNPs flagged with the conditions following the GATK best practices: QD < 2.0, FS > 60.0, SOR > 3.0, MQ < 40.0, MQRankSum < −12.50, ReadPosRankSum < −8.0. We ended up with a filtered call set of 16,316,814 biallelic SNPs and 250 accessions.

We produced per-individual and per-locus missingness statistics with vcftools v.0.1.16[59]. We used vcftools—mindDP to mask genotypes with a <2 depth. Then we minimised the extent of missing data due to reference genome bias by filtering out SNPs that were missing in >5% of the individuals and kept individuals with missing data <0.40. We therefore ended up with 1,910,119 biallelic SNPs across 247 individuals. A stringent threshold for the missing rate per locus was set to minimise missing data by individuals and to focus mainly on SNPs shared by the four species, reflecting their ancient histories. However, to check consistency of our stringent threshold used for the per-locus missing rate, we generated two other SNP datasets with less stringent thresholds at 10% and 20%.

### k-mer analysis
We used the k-mers approach to analyse genetic relationships without mapping bias. We generated a k-mers presence/absence table from cleaned fastq reads. We used the KMERS_GWAS step[60] from the iKISS pipeline (https://forge.ird.fr/diade/iKISS) to convert reads into k-mers. We decomposed the reads of each individual into 31-sized k-mers. We filtered out k-mers that were not in canonised form (k-mers with reverse complements) in at least 20% of individuals. We also only kept k-mers with a minor allele count ≥ 2 and a minor allele frequency > 0.05. We generated a 1,000,000 k-mer presence/absence table and converted it with vcftools into a plink bed/ped file for population genomic analyses.

### Population diversity statistics
We computed the number of segregating sites (S), the nucleotide diversity (π), Watterson's Θ ($Θ_w$) and Tajima's D per species with egglib v.3.2.1[61], which is a Python module that enables fast computation of population genomic diversity statistics. Statistics were computed across the genomes using sliding windows of size 50 kb and step sizes of 10 kb. A threshold of 20% for the per-locus missing rate was applied for each species separately. LD decay ($r^2$) was computed per species with PopLDdecay v.3.42[62] within a 500 kb distance. We then used the Perl script *Plot_MultiPop.pl* and the ggplot2 v.3.4.2[63] package of R to plot the figures.

The Jaccard index (JI) was calculated with the k-mer dataset as a measure of genetic diversity within species without mapping bias. We estimated a dissimilarity index based on the proportion of shared and non-shared k-mers from the 1,000,000 presence/absence table, which was split into 100 tables of random samples of 10,000 k-mers from which the JI values were obtained.

## Population genetic structure

All datasets used for the following analyses were filtered for minor allele frequency (MAF) at 0.05. We performed PCA analyses on SNP and k-mer datasets with the pca function from the LEA v.3.9.5 R package[64–66]. Model-based clustering of genotypic data was performed with the sparse nonnegative matrix factorisation method (sNMF) implemented in the LEA v.3.9.5 R package[64–66]. We ran 10 independent runs per $K$, with $K$ ranging from 2 to 14. The optimal $K$, indicating the number of ancestral populations that best explained our data, was determined with the cross-validation error rates. We displayed the respective barplots with the pophelper v.2.3.1 R package[67]. For all inferred pairs of genetic clusters, we computed the pairwise population distances $dxy$ and the net nucleotide divergence $da$ with egglib v.3.2.1[61].

## Historical relationships between populations

We first inferred between-population relationships with Treemix v.1.13[27]. Treemix v.1.13[27] uses the genome-wide covariance structure of allele frequencies between populations to maximise the likelihood of the inferred tree topology and migration between populations. We first analysed the four groups together, i.e. the two cultivated species *D. iburua* and *D. exilis*, and the two wild species *D. longiflora* and *D. ternata*. We also performed an analysis based on the genetic structure inferred with sNMF. Individuals were included in a group if their ancestry coefficients were 0.6 or higher. First, a total of 100 independent bootstrap replicates were generated and a consensus tree was built with phylip v.3.697. We then ran 10 independent Treemix runs with the consensus tree to find the optimal number of migration events ($m$), for each m from $m = 1$ to $m = 6$. The optimal number of migration edges was determined with the optM v0.1.6 R package[68] which allowed us to plot the mean composite log likelihood of each run for each $m$. Another set of 100 bootstrap replicates was generated to find the new consensus tree with the number of previously inferred migration edges. Finally, 30 independent Treemix runs were performed with the new consensus tree and we visualised the tree having the maximum likelihood with R Treemix plotting functions. At each step of this pipeline, we accounted for linkage disequilibrium by setting SNP block sizes ranging from 500 to 1000. The model was implemented without specifying a root or by specifying *D. ternata* individuals from Côte d'Ivoire as root, since they appeared to be distant and isolated in the PCA and clustering analyses. This pipeline was implemented based on an available approach (https://github.com/carolindahms/TreeMix). In addition to this genome-wide approach, we summarised relationships among the different populations with a sliding window method implemented in the Twisst pipeline[28] which quantifies the relative weights for each possible subtree topology among a set of inferred gene trees (Supplementary Method 2).

## Inference of effective population size histories

To investigate the demographic history of each cultivated fonio millet species and wild relatives, we inferred past changes in effective population size of our population with a sequentially Markovian coalescent based approach implemented in the smc++ v.1.15.4 software package[32]. We first implemented new variant calling procedures from previously aligned sequences to obtain specific and restricted VCF files per target population. We used the bcftools mpileup (v.1.16) algorithm[57] associated with the *bamcaller.py* python script from msmc-tools (https://github.com/stschiff/msmc-tools) to output vcf files with genotype likelihood probabilities, and to mask files containing sites and intervals having a sequencing depth greater than (mean depth)/2 and less than (mean depth) × 2. The mean depth was computed separately for each dataset used for inference. We used the bedtools v.2.30.0[69] complement command to obtain mask files containing positions to be labelled as missing data (across all samples) for smc++ inferences, corresponding to insufficiently covered regions. Smc++

input files were then generated with the vcf2smc script with the --mask option.

Different sets of distinguished lineages were considered for the different sets of individuals (all samples were considered as distinguished) when outputting the smc.gz input files for the inferences. Finally, we used the estimate command with a $6.5 \times 10^{-9}$ mutation rate to infer past population effective sizes, with 30 independent runs. The results were plotted with the ggplot2 v.3.4.2 R package[63]. We considered a 1 year generation time. The analysis was conducted for different sets of populations considering species delimitations and genetic clusters.

## Population modelling and estimation of the relative timing of fonio millet domestication

We also assessed the relationship between the two cultivated fonio species and the two wild species using a maximum composite likelihood approach implemented in fastsimcoal v.2.8[29–31]. We tested different models corresponding to the likelihood of different coalescent topologies to investigate, with another methodology, the best supporting topology for the cultivated fonio millets and the two wild relative species.

We defined the four populations according to the species and population genetic structure analyses. We excluded *D. ternata* individuals from Côte d'Ivoire and the *D. longiflora* East African group from the analyses, as both were more genetically differentiated from the cultivated gene pools than the other wild individuals. We considered one population each for *D. exilis* and *D. iburua*, excluding admixed individuals with a membership coefficient <0.6. We subsampled the 203 *D. exilis* accessions to obtain 21 accessions representative of the species diversity and geographic distribution. This subsampling was done to avoid major bias in sample size differences with *D. iburua*. We thus retained 54 individuals and four populations for coalescent simulations: six for *D. ternata*, eight for *D. longiflora*, 21 for *D. exilis* and 19 for *D. iburua*.

We calculated the site frequency spectrum (SFS) based on the genotype likelihoods of each population with ANGSD v.0.940[70], which relies directly on the bam files. First, site allele frequency likelihood files (.saf) were generated with the samtools genotype likelihood model (-GL 1 option). We kept sites present in at least 80% of the individuals and filtered out poorly aligned reads and base quality (--minMapQ 30 --baq 1). We then computed the folded joint frequency spectrum of each population pair with realSFS from ANGSD v.0.940[70], which served as input to fastsimcoal to compare the observed and estimated log-likelihoods of the simulated models.

We first tested six different topologies (Supplementary Fig. 18) to test: (1) the independence of the cultivated fonio millet domestications, and (2) from which wild species the cultivated species differed. We assumed a constant population size for these different scenarios. The best tree topology was then used as a backbone to test if the data supported or not a domestication bottleneck after population divergence.

For each of the six scenarios, we performed 100 independent runs of 500,000 coalescent simulations to estimate: the effective size of each population X ($N_X$) and the times of population divergence ($T_{X-Y}$). We defined uniform (or log-uniform) prior distributions as well as prior boundaries per parameter (Supplementary Table 6). Parameter estimations were based on a maximum composite likelihood from the SFS with 40 optimisation cycles, with parameter reinitialization after three non-improved cycles while reducing the search ranges by 50%. We used an Infinite site model (-I) with a fixed mutation rate of $6.5 \times 10^{-9}$ and a 1 year generation time as fonio millets are annual plants.

According to the authors' recommendations[30], the final maximum likelihood estimation was based on 100 new runs using the estimated point values of parameters inferred by the previously obtained best run. Models were compared using the likelihood distribution of the models and the Akaike information criterion.

Once we determined the most likely fonio millet domestication scenario, we estimated confidence intervals of the parameter estimates through a parametric bootstrap approach. Using the best set of parameters, we generated 500,000 SFS from 100 independent pseudo-observed datasets of 200,000 non-recombining DNA segments of 1000 bp. We set the initial parameter estimation values with the -initvalues command line.

## Reporting summary

Further information on research design is available in the Nature Portfolio Reporting Summary linked to this article.

## Data availability

White and black fonio accessions are conserved in national collections, and duplicates covered by the Material Transfer Agreement (MTA) are stored in the ARCAD gene bank (Montpellier, France). The raw sequences retrieved from Abrouk et al.[23] are available on EBI-ENA under accession PRJEB36539. The annotation of the CM05836 genome is available at the DRYAD database [https://doi.org/10.5061/dryad.2v6wwpzj0]. The new raw sequencing data re-sequenced in this study are available at EBI-ENA under accession PRJEB80862. Passport data of the accessions and VCF files used to perform the analyses are openly available in DataSuds repository [https://doi.org/10.18167/DVN1/OYTQO6]. Data reuse is granted under CC-BY licence. Source data are provided with this paper.

## Code availability

Scripts for the bioinformatics pipeline and for the different analyses carried out throughout the paper are available in the Zenodo repository [https://zenodo.org/records/15267221]. Previously reported pipeline and codes to run the k-mer analyses are available on the IRD Forge [https://forge.ird.fr/diade/iKISS].

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

## Acknowledgements

We thank all local communities and authorities for their collaboration and participation. Collection and genetic analysis of fonio landrace diversity over the last decade have been supported through several research projects funded by WAAPP/PPAAO 2A (CERA58ID06 SE), Agropolis Fondation (Agropolis Resource Center for Crop Adaptation and Diversity [ARCAD] project and the Cultivar project—ID 1504-007) through the Investissements d'Avenir programme (Labex Agro: ANR-10-LABX-0001-01) within the framework of I-SITE MUSE (ANR-16-IDEX-0006), by the French national ANR project (AfriCrop project, ANR-13-BSV7-0017) and by the European Union Horizon 2020 research and innovation programme (EWA-BELT, 862848, 'Linking East and West African farming systems experience into a BELT of sustainable intensification'). We are grateful to the ALF CIRAD herbarium and the herbarium of the Muséum National d'Histoire Naturelle (MNHN) in Paris for providing the *D. ternata* specimens. The bioinformatics analyses were performed on the Core Cluster of the Institut Français de Bio-informatique (IFB) (ANR-11-INBS-0013).

## Author contributions

T.K., P.C., Y.V., A.B., C.B. and C.L. designed research. E.A.U., H.O.O., S.N.D., C.O.A.A., E.S., E.G.A.D., A.R.I.B.Y., S.I.S., Y.B., B.M.D., M.C.G., R.Y.A., J.A.D., M.G., J.J.W., S.C., T.K. and C.B. contributed to the sampling of the biological material. M.C. and S.C. performed molecular analyses and constructed the sequencing libraries. T.K. and P.C. established the SNP calling pipeline and T.K. performed bioinformatics analyses. T.K. and J.O. performed the k-mer analysis. T.K. analysed the data with contributions from P.C., J.O., C.L., A.B., C.B. and Y.V. T.K. wrote the paper with substantial inputs from P.C., L.C., Y.V., A.B., C.B. and C.L. All authors have read and approved the manuscript.

## Competing interests

The authors declare no competing interests.

## Additional information

[1]CIRAD, UMR AGAP Institut, Montpellier, France. [2]AGAP Institut, University of Montpellier, CIRAD, INRAE, Institut Agro, Montpellier, France. [3]DIADE, University of Montpellier, IRD, CIRAD, Montpellier, France. [4]Department of Genetics and Biotechnology, University of Calabar, Calabar, Nigeria. [5]Center for Crop Improvement, Nutrition & Climate Change (CCINCC), Ebonyi State University, Abakaliki, Nigeria. [6]Department of Crop Production, Faculty of Agriculture, University of Jos, Jos, Plateau State, Nigeria. [7]Genetics, Biotechnology and Seed Science Unit (GBioS), Laboratory of Plant Production, Physiology and Plant Breeding (PAGEV), School of Plant Sciences, University of Abomey-Calavi, Abomey-Calavi, Cotonou, Republic of Benin. [8]LaPAPP, University of Parakou, Parakou, Republic of Benin. [9]Department of Rainfed Crop Production (DCP), National Institute of Agronomic Research of Niger (INRAN), Niamey, Niger. [10]Department of Plant Production and Biodiversity, Faculty of Agronomic and Ecologic Sciences, University of Diffa, Diffa, Niger. [11]Laboratory for the Management and Valorization of Biodiversity in the Sahel (GeVaBioS), Abdou Moumouni Univyersit, Niamey, Niger. [12]Department of Biology, Faculty of Science and Technic, Abdou Moumouni University, Niamey, Niger. [13]Institut Sénégalais de Recherches Agricoles (ISRA), Centre d'Etude Régional pour l'Amélioration de l'Adaptation à la Sécheresse (CERAAS), Thiés, Sénégal. [14]Council for Scientific and Industrial Research—Savanna Agricultural Research Institute (CSIR-SARI), Nyankpala, Ghana. [15]Laboratoire de Botanique, Département de Botanique et Géologie, IFAN Ch. A. Diop/UCAD, IRL 3189 « Environnement, Santé et Société », Université Cheikh Anta Diop, Dakar, Sénégal. [16]Naturalis Biodiversity Center, Leiden, The Netherlands. [17]These authors contributed equally: Yves Vigouroux, Claire Billot, Adeline Barnaud, Christian Leclerc. ✉e-mail: thomas.kaczma@gmail.com; yves.vigouroux@ird.fr; claire.billot@cirad.fr; adeline.barnaud@ird.fr; christian.leclerc@cirad.fr

