## [Peer review file · Nature Communications]

Independent domestication and cultivation histories of two West African indigenous fonio millet crops

Corresponding Author: Dr Thomas Kaczmarek

Version 0:

Reviewer comments:

Reviewer #1

(Remarks to the Author)

This paper is well written with proper and comprehensive statistical analyses. It is an important advancement for this important African orphan crop. The paper provides new genomes for fonio, diversity in 265 accessions with passport data and a convincing analyses using multiple sources of evidence of its domestication and ethnobotany. The data and analyses clearly support the conclusions with detailed methods.

Specifically

All Excel tables should have data separated into columns rather than collated in a single cells

Ln 77. Enoch Achigan-Dako, pers. Comm can be dropped as Achigan-Dako is author

Ln 100, switch individuals to accessions.

Ln 104. D. exilis were derived

Ln 106. largest public collection. Generally avoid statements using “largest, best, top etc.” They can be challenged or incorrect even before the paper is published.

Ln122-123. Is © references ok format for journal?

Ln 128—same comment as for line 106 with “for first time”—drop.

Ln 137 switch missingness to “proportion missing” or “missing rate” throughout.

Ln 153-155. It is not clear how robustness was measured or its basis when comparing data form different missing rates. Explain or delete.

Ln 298. Not sure that data suggests that black and white fonio should be separated for breeding purposes—perhaps expand this idea due to their significantly different genetic backgrounds,

Ln 304. .. but white and black fonio are not separated³³.

Ln 407—again—how do you know its geographical region is fully represented—I suggest using “is likely fully represented”

(Remarks on code availability)

Reviewer #2

(Remarks to the Author)

This manuscript provides new and interesting data on the diversity and domestication of two important but overlooked African crops. I am surprised the authors are confident in assessing domestication histories and timings with only one wild relative included for each crop species, especially as the introduction admits the lack of universal agreement on the closest relatives. This work would be stronger with an assessment of all the local and morphologically similar taxa.

(Remarks on code availability)

Reviewer #3

(Remarks to the Author)

In "Independent domestication and cultivation histories of two West African indigenous fonio millet crops" Kaczmarek and colleagues study the domestication history of black and white fonio and two wild relatives.

The study of the domestication history of crops is important and can give insight into crop and human history due to their close relationship. African crops have historically received less attention, but research on them has caught up in recent years.

The study employs standard population genetic methods to identify the domestication history of the two fonio crops. For this study 94 new genomes were re-sequenced out of which 17 (18%) were removed due to low data quality in addition 157 public genomes have been added. While the findings are interesting, they represent a rather common case of multiple domestication that however is not closer examined.

The history of white fonio has previously been reconstructed (Abrouk et al 2020 Nat. Com). The relationships between the 4 species seems to have been previously resolved and are confirmed now with genomic data.

While the results seem robust and interesting for fonio research, I am not sure about the broader implications of the findings. Neither methodologically, given the rather basic population genetic analysis nor from the results as these seem rather crop specific. The discussion in the light of human history is interesting and well integrated. I cannot judge well to what degree the archaeological aspects are novel.

Additional points:

- strong imbalance between domesticated and wild accessions. maybe it would be worth subsampling the white fonio samples for the PCA etc, as they might bias the analysis due to their high number compared to the others.

- what can explain the extreme difference in genetic diversity? Might this be a result of reference bias during SNP calling?

- I think a more cautious discussion of the demographic analysis would be beneficial. SMC++ has no power to infer information about recent history (e.g. 500 years). In addition the timing is highly dependent on the mutation rate estimate. No justification for the used mutation rate estimate is being given, but I doubt that this has been measured in fonio.

(Remarks on code availability)

I only looked up if code is available. There is a very nice repository with readme. I did not examine the code for correctness or reproducibility.

Version 1:

Reviewer comments:

Reviewer #1

(Remarks to the Author)

Thank you for addressing all comments sufficiently. No more edits.

(Remarks on code availability)

Reviewer #2

(Remarks to the Author)

Thank you for addressing my concerns in detail.

(Remarks on code availability)

N/A

Reviewer #3

(Remarks to the Author)

Most of my comments have been addressed and improvement have been made.

One essential concern regarding SNP calling and filtering remains, as this would strongly influence the inferred results.

Regarding the statement:

"The use of the *D. exilis* reference genome undoubtedly biased our estimates of genetic diversity. Furthermore, as *D. ternata* and *D. iburua* did not map well on this reference genome, we used a stringent threshold (5%) on the locus missing rate to minimise missing data among individuals. This resulted in eliminating intraspecific diversity in *D. exilis* and *D. longiflora*, i.e. the SNPs present in these species but not in *D. iburua* and *D. ternata*.

Supplementary Table 2 indicates that using a less stringent threshold on the locus missing rate (20%) increased the estimates of genetic diversity π for *D. exilis* and *D. longiflora*, thus reducing the difference in genetic diversity between the two pairs of species. "

Filtering strong on missing SNPs when one species does not map well, will lead to even stronger bias. Particularly as the number of samples in the wild species is very low. For instance with your 5% filter across all samples an SNP with an allele frequency of 50% (which is high) in the wild population, but absence in the reference population will be removed and not used to calculate diversity. Filtering for diversity measures should be considered within species. In this light it is actually complicated to understand how samples with N=11 can have such high diversity. It might well explain the low differentiation between species though.

It's probably worth looking at allelic richness and the number of private alleles per species.

The k-mer approach might be helpful, and it might make sense to exchange the tables in the main text.

(Remarks on code availability)

Version 2:

Reviewer comments:

Reviewer #3

(Remarks to the Author)

Thank you for the additional clarifications.

(Remarks on code availability)

Response to reviewers

- Referees' comments are shown in black
- Our responses to the referees' comments are italicized in grey.
- We provide citations from the manuscript, with our changes shown in red.

Reviewer #1 (Remarks to the Author):

This paper is well written with proper and comprehensive statistical analyses. It is an important advancement for this important African orphan crop. The paper provides new genomes for fonio, diversity in 265 accessions with passport data and a convincing analyses using multiple sources of evidence of its domestication and ethnobotany. The data and analyses clearly support the conclusions with detailed methods.

We appreciate your positive feedback on our research.

Specifically

All Excel tables should have data separated into columns rather than collated in a single cells

Corrected. The format of the three tables provided were updated to .txt (tab separated).

Ln 77. Enoch Achigan-Dako, pers. Comm can be dropped as Achigan-Dako is author

Corrected.

Ln 100, switch individuals to accessions.

Corrected.

Ln 104. D. exilis were derived

Corrected.

Ln 106. largest public collection. Generally avoid statements using “largest, best, top etc.” They can be challenged or incorrect even before the paper is published.

*Corrected. This sentence was changed to: “We assembled **a large** collection of genomic resources comprising 265 accessions of the four species.”*

Ln122-123. Is © references ok format for journal?

We have noted this copyright symbol in articles published in Nature Communications, so we think it's OK.

Ln 128—same comment as for line 106 with “for first time”—drop.

Deleted.

Ln 137 switch missingness to “proportion missing” or “missing rate” throughout.

Corrected. We switched missingness to “missing rate” throughout and in the legend of Fig. 2.

We also changed the sentence lines 163-164: “The results obtained with the SNP datasets that had been filtered for locus missingness at different missing data thresholds...”.

For clarity, we added the per- prefix in line 465: “We produced per-individual and per-locus missingness statistics with vcftools v.0.1.16⁵⁷.”

The sentence line 469 was changed to “A stringent threshold for the missing rate per locus missingness threshold was set to minimize missing data by individuals and...”

The sentence line 486 was changed to: “Statistics were computed with the three SNP datasets filtered at different thresholds for the missing rate per locus locus missingness (5%, 10% and 20%).”

Ln 153-155. It is not clear how robustness was measured or its basis when comparing data form different missing rates. Explain or delete.

*Thank you for this point. We have modified the sentence to make it more explicit: “With less stringent thresholds for the locus missing rate (10% and 20% instead of 5%), the genetic diversity estimates were still higher for *D. iburua* and *D. ternata* than for *D. exilis* and *D. longiflora*. The Tajima's *D* values were still negative for the *D. exilis*/*D. longiflora* pair and positive for the *D. iburua*/*D. ternata* pair (Supplementary Table 2).*

*We suppressed: Our result was robust when performing analyses with higher SNP missing rates (10% and 20% instead of 5%, Supplementary Table 2). A too stringent missing rate of 5% excluded polymorphic sites associated with *D. exilis* and *D. longiflora*.”*

Ln 298. Not sure that data suggests that black and white fonio should be separated for breeding purposes—perhaps expand this idea due to their significantly different genetic backgrounds,

You are right, clarification was needed here. The idea is that white and black fonio are genetically very different and therefore form two distinct genetic entities on which conservation and crop improvement programmes should be focused.

We have thus modified the text accordingly: “Our results highlighted the genetic uniqueness of each fonio millet, which have two distinct gene pools. This information is crucial for conservation and crop improvement programmes as each species requires tailored strategies to ensure its effective use in agrosystems. It also presents an opportunity for breeding programmes, as key traits such as plant architecture differ between white and black fonio, potentially offering key avenues for crop improvement.”

The former sentence was: “~~Our results highlighted that white and black fonio should be considered separately for conservation and breeding purposes.~~”

Ln 304. “.. but white and black fonio are not separated³³.”

We have not modified the text here because we really meant: “black and white fonio are not included”.

Ln 407—again—how do you know its geographical region is fully represented—I suggest using “is likely fully represented”

Corrected.

Reviewer #2 (Remarks to the Author):

This manuscript provides new and interesting data on the diversity and domestication of two important but overlooked African crops. I am surprised the authors are confident in assessing domestication histories and timings with only one wild relative included for each crop species, especially as the introduction admits the lack of universal agreement on the closest relatives. This work would be stronger with an assessment of all the local and morphologically similar taxa.

Thank you for finding our research interesting and important for these two crops that have been overlooked in mainstream research.

Our study aimed to clarify the genetic and evolutionary relationships between the cultivated species. The fundamental question addressed in our article is: Were white and black fonio domesticated independently?

Previous studies have found some evidence of genetic differentiation between the two fonio millets using AFLP, RAPD or SSR markers, yet no authors have jointly considered the genetic

diversity and geographical distribution of cultivated species and their closest wild relatives. This has made it hard to draw confident conclusions on the independence of the domestication history of both fonio millets, and about the extent of gene flow between them.

*We considered it appropriate to address these issues and answer the main question of our study using the acknowledged closest wild relatives. Morphological and molecular data show that *D. longiflora* and *D. ternata* are the main contributors to the genomes of cultivated fonio millets. We are thus confident that *D. longiflora* and *D. ternata* are the closest wild relatives of white and black fonio, respectively. Given our genomic analyses and results, we can affirm that white and black fonio were domesticated independently, without gene flow between the two pairs of cultivated/wild species.*

*Our study did not aim to construct a phylogeny of several other *Digitaria* wild and cultivated taxa. However, yes indeed, future studies with more wild taxa could enhance our understanding of the *Digitaria* phylogeny. We hope that our study will promote further research based on genomic data on other wild *Digitaria* species.*

Reviewer #3 (Remarks to the Author):

In "Independent domestication and cultivation histories of two West African indigenous fonio millet crops" Kaczmarek and colleagues study the domestication history of black and white fonio and two wild relatives.

The study of the domestication history of crops is important and can give insight into crop and human history due to their close relationship. African crops have historically received less attention, but research on them has caught up in recent years.

The study employs standard population genetic methods to identify the domestication history of the two fonio crops. For this study 94 new genomes were re-sequenced out of which 17 (18%) were removed due to low data quality in addition 157 public genomes have been added. While the findings are interesting, they represent a rather common case of multiple domestication that however is not closer examined.

We thank you for highlighting the importance of studying the history of the domestication of crops from both fundamental and applied standpoints. The history of white and black fonio was not completely clear, i.e. did it involve 'multiple domestication' of a single wild species,

or independent domestication of two different species? We generally used ‘multiple domestication’ when a single species has been domesticated more than once.

In our study, we provided clear evidence that white and black fonio are two genetically distinct species, and that they resulted from two independent domestication events, not multiple events (e.g. in apricots (*Prunus armeniaca*) or cabbage (*Brassica oleracea*)). We could draw a parallel with two species known as “rice”, i.e. African rice (*Oryza glaberrima*) and Asian rice (*Oryza sativa*). The case of fonio millets is all the more remarkable because, unlike rice, which was domesticated independently on two different continents, the domestication of fonio millets occurred in the same region of Africa. This raises several new research questions regarding the spatiotemporal scales of domestication, as well as the ethnic groups involved in the process. Further in-depth collaboration with archaeobotanical specialists would be essential to address these questions and enhance our understanding of the domestication history of fonio.

Spurred by your remark, we decided to clarify this message in the discussion in the manuscript. We changed the first sentences of the section “Independent domestication events for the two fonio millet species” as follows (lines 295-301):

*“The domestication of plants and animals can have single or multiple origins^{2,38,39}. Apricots (*Prunus armeniaca*), for instance, were domesticated from two distinct wild populations that exchanged genes². On the other hand, *Oryza glaberrima* and *Oryza sativa*, or so-called African and Asian ‘rice’, respectively, resulted from independent domestication events on two different continents⁴⁰. The fonio millet case is all the more remarkable because, unlike rice, the domestication of black and white fonio millets occurred in the same region of Africa, with no gene flow between them.”*

~~The old sentences were: Many similar crops or animals of the same species have been domesticated more than once^{2,37,38}. Recent research has also highlighted the complex relationship between domestication in different species, with one of the most controversial being Asian rice³⁹. Our study statistically confirmed the independence of the domestication of the two cultivated fonio species, with no gene flow between them.”~~

The history of white fonio has previously been reconstructed (Abrouk et al 2020 Nat. Com).

Abrouk et al. (2020) only focused on white fonio and provided a high-quality genome assembly of *D. exilis* while analysing the genetic diversity of 157 white fonio accessions,

along with 14 accessions of its D. longiflora wild relative. We accounted for more accessions than those covered in this initial study, notably in key regions (Senegal, Niger), but more importantly in Nigeria where white and black fonio co-occurred. This initial study did not study relationships between black and white fonio, nor the probability of gene flow. We reached beyond the current knowledge and used a new modelling approach to shed light on the timing of expansion inferred using smc++ (Fig. 3d, Supplementary Fig. 17). Our results combined with new recent archaeological research findings regarding fonio provide strong evidence of an initial spread of fonio from Nigeria at the beginning of the CE era—a question Abrouk et al. 2020 did not address.

The relationships between the 4 species seems to have been previously resolved and are confirmed now with genomic data.

You are right that previous studies seem to have resolved the relationships between the four species. However, previous results have been partial because they have not taken the genetic diversity and geographical distribution of cultivated species and their closest wild relatives jointly into account.

The novelty of our study is that we robustly assessed the question of the independence of the domestication of the two fonio millets by integrating the four species together, in particular white and black fonio individuals from the same localities, which could have exchanged gene flow during the domestication process. The findings of all methods we used, whether using a reference genome or not, provided evidence in favor of two independent domestication events, without gene flow.

While the results seem robust and interesting for fonio research, I am not sure about the broader implications of the findings. Neither methodologically, given the rather basic population genetic analysis nor from the results as these seem rather crop specific.

Thank you for highlighting the robustness of our results and their importance for fonio research. Indeed, fonio millet comprises two orphan crops which so far have been overlooked by research. We were delighted to see a Nature collection on “Orphan crop genomics and improvement” as we are convinced that research should focus on these locally adapted crops. Prior to this study, no full genomes of D. iburua were available. Given the high genetic diversity of this species and the limited cultivation area, our results are of substantial importance for its conservation and further improvement. Indeed, this species could be crucial for the adaptation of agricultural systems to climate change.

In addition to common population genomics methods, we have shown that k-mers are particularly useful for studying the diversity of orphan crops, which often lack a reference genome. We therefore encourage the use of k-mers in genomic studies of neglected crops. We have highlighted this in the revised version of the manuscript (lines 254-262):

*“The findings obtained via the k-mer approach, which directly relies on the sequencing reads, were particularly relevant as *D. iburua* and *D. ternata* did not map well on the *D. exilis* reference genome. This mapping-free approach served as a validation criterion for using the SNP dataset produced with the available reference genome²³. Congruent results were obtained with both approaches regardless of the use of PCA analyses and genetic clustering methods. **We therefore recommend using k-mers as a validation criterion in further genomic studies on indigenous crops (which often lack a reference genome), but also as a means to study their genetic diversity.**”*

We would also like to further stress the implications of the findings, which we believe are not limited to fonio research. First, our findings should stimulate interest in studying the genetic diversity and evolutionary history of the numerous other neglected and underutilised crops on all continents. The acquisition of fundamental knowledge on a wide range of indigenous crops will facilitate their use in breeding programmes for agricultural diversification.

Secondly, a broader implication is the importance of these results for our knowledge of plant domestication, especially the origins of agriculture in sub-Saharan Africa. In this paper, with two other species, we highlighted that the domestication of African crops took place in a much more restricted area than postulated by Harlan (1976). The domestication non-centre hypothesis had already been ruled out by studies on pearl millet (Burgarella et al., 2018), African rice (Cubry et al., 2018) and yam (Scarcelli et al., 2019), three crops domesticated at different regions near the Niger River. Our results, combined with archaeological data, suggest that white and black fonio domestication occurred somewhere around Nigeria, i.e. further east than the three crops mentioned above. Our study therefore paves the way for a better understanding of the spatiotemporal nature of crop domestication in Africa and the shared history of humans and crops.

*We decided to add a paragraph to the discussion to highlight this impact of our work (lines 364-372): **“These results enhance our knowledge on the origins of agriculture in sub-Saharan Africa and support the view that the domestication of African crops took place in a much more restricted area than postulated by Harlan⁴⁷. The domestication non-centre hypothesis***

had already been ruled out by the findings of genomic studies on pearl millet⁴¹, African rice⁴², and yam⁴³, i.e. three crops domesticated at three different locations near the Niger River, thereby constituting a West African cradle of domestication. Here we may add a fourth location, around Nigeria, from which fonio may have originated. Our study therefore paves the way for gaining greater insight into the spatiotemporal nature of crop domestication in Africa and the shared history between humans and crops.”

The discussion in the light of human history is interesting and well integrated. I cannot judge well to what degree the archaeological aspects are novel.

Thank you for your comment. Regarding the archaeological data, we reviewed the spatiotemporal fonio archaeobotanical evidence. Fonio had previously been identified in two geographically contrasted sites (Cubalel in Senegal and Janruwa C in Nigeria) during the same period. The samples discovered in Cubalel actually did not correspond to fonio. We have therefore highlighted that the oldest archaeobotanical samples of fonio originated from a single site, in Nigeria, dating back to the first centuries AD. This suggests a primary center of fonio domestication in Nigeria.

Additional points:

- strong imbalance between domesticated and wild accessions. maybe it would be worth subsampling the white fonio samples for the PCA etc, as they might bias the analysis due to their high number compared to the others.

*We agree that subsampling is relevant for some analyses with unbalanced datasets. We used subsamples in our study, particularly for demographic analysis and scenario modelling. For example, in fastsimcoal simulations, we subsampled the 203 *D. exilis* accessions to obtain 21 accessions representative of the species diversity and geographic distribution, and thus had a more balanced dataset. We also subsampled *D. exilis* using the twisst method (Supplementary Text 3, Fig. 3b, Supplementary Fig. 14), and the results were consistent with those obtained with the TreeMix method (Fig. 3a, Supplementary Fig. 13).*

Based on your remark, we performed PCA with SNP and KMER datasets for the 21 white fonio accessions. The figures obtained are presented below. For SNPs, the first axis differentiated the two pairs of cultivated and wild species, while the third axis differentiated

D. longiflora from *D. exilis*. The three *D. ternata* from Côte d'Ivoire were clearly discriminated on the second axis.

Figure R1. Principal component analysis focused on the SNP matrix with a subsample of 21 *D. exilis*. Left: first and second axes displayed. Right: first and third axes displayed.

The same pattern of differentiation was obtained with kmers (Figure R2), whether using all samples or reduced sampling.

Figure R2. Principal component analysis performed with kmers, with a subsample of 21 *D. exilis*. First and second axes displayed.

Thus, these results confirmed those obtained with the full dataset used in our study. This is not

surprising as PCA is much more sensitive to outliers than to clustered data. We think that we should keep the figures and results as they are presented in the manuscript.

- what can explain the extreme difference in genetic diversity? Might this be a result of reference bias during SNP calling?

*The use of the *D. exilis* reference genome undoubtedly biased our estimates of genetic diversity. Furthermore, as *D. ternata* and *D. iburua* did not map well on this reference genome, we used a stringent threshold (5%) on the locus missing rate to minimise missing data among individuals. This resulted in eliminating intraspecific diversity in *D. exilis* and *D. longiflora*, i.e. the SNPs present in these species but not in *D. iburua* and *D. ternata*. Supplementary Table 2 indicates that using a less stringent threshold on the locus missing rate (20%) increased the estimates of genetic diversity π for *D. exilis* and *D. longiflora*, thus reducing the difference in genetic diversity between the two pairs of species.*

*Based on your comment, we decided to perform some genetic diversity estimations using kmers. We used the Jaccard dissimilarity index statistic to assess the diversity within each species (see Ondov et al., 2016). *D. iburua* and *D. ternata* still showed higher diversity than *D. exilis* and *D. longiflora* (Figure R3, preliminary results). Hence, trends in the SNP-based results were in line with these new analysis findings.*

Differences in reproductive systems may also contribute to this difference in genetic diversity.

*These points are highlighted in the discussion, with some changes (in red) for clarification (lines 284-293): “The genetic diversity of black fonio (*D. iburua*) and its wild relative (*D. ternata*) was **much** higher than for white fonio (*D. exilis*) and *D. longiflora*. This was probably due to the use of the *D. exilis* reference genome which differs markedly from the *D. iburua* and *D. ternata* genomes, or it could reflect differences in reproduction systems among the two species pairs. In fact, the self-fertilization rate of *D. exilis* is around 99%³⁵. Outbreeding might actually be more pronounced for black fonio and its wild relative, thus increasing their genetic diversity. **The genetic diversity of populations and species could potentially be studied using indices calculated with kmers, such as the Jaccard dissimilarity index³⁶, which allows calculation of the proportion of shared kmers within a population. Such approaches are receiving increasing attention in the of population genomics field when no reference genome is available³⁷.”***

Figure R3. Preliminary results on genetic diversity within species calculated using Jaccard Average dissimilarity. This index reflects the proportion of non-shared kmers within the population.

- I think a more cautious discussion of the demographic analysis would be beneficial. SMC++ has no power to infer information about recent history (e.g. 500 years). In addition the timing is highly dependent on the mutation rate estimate. No justification for the used mutation rate estimate is being given, but I doubt that this has been measured in fonio.

Thanks for your comment, we have amended the discussion.

Indeed, inferring demographic changes in the recent past using coalescent methods can be unreliable. However, SMC++ is the most relevant method for inferring recent demographic changes (e.g. 500 years). Compared to other methods like PSMC or MSMC, the use of a larger number of genomes in SMC++ increases the probability of having coalescent events in the recent history of the population. Consequently, it makes sense to detect events < 500 generations old. But we agree that it is essential to always be cautious in interpreting data, particularly changes in population size and the timing of these changes. For example, it is known that population structure can affect this type of analysis. Otherwise, the mutation rate has not been measured in fonio, so we used one based on grasses.

We have therefore amended the discussion to take these biases into account (lines 379-391):

“Variations in effective population size should be interpreted with caution as the population structure can lead to incorrect inferences⁴⁸. The inferred times for bottlenecks or population expansion also depend on the mutation rate, which has not yet been determined for fonio. In addition, the power to estimate effective population sizes decreases as the time becomes more

recent, but coalescent methods using several haplotypes (from a few dozens to hundreds) provide higher resolution than other methods for inferring changes in population size in the recent past³². ~~More recently,~~ D. exilis experienced a decline in effective population size around 500 years ago, which was also the pattern we noted for D. iburua in some smc++ runs. This decline has also been observed in similar studies for African yam⁴³ (Dioscorea rotundata) and African rice (Oryza glaberrima)⁴². With fonio, there are now three different signs of a recent loss of indigenous African crops, which could be ~~and was~~ associated with major social and agricultural changes, intensification of the slave trade, and the introduction of new, less labour-intensive crops such as maize, cassava and Asian rice.”.

Reviewer #3 (Remarks on code availability):

I only looked up if code is available. There is a very nice repository with readme. I did not examine the code for correctness or reproducibility.

Thank you for your attention to this repository.

Reviewer #3 (Remarks to the Author):

Most of my comments have been addressed and improvement have been made. One essential concern regarding SNP calling and filtering remains, as this would strongly influence the inferred results.

Regarding the statement:

"The use of the *D. exilis* reference genome undoubtedly biased our estimates of genetic diversity. Furthermore, as *D. ternata* and *D. iburua* did not map well on this reference genome, we used a stringent threshold (5%) on the locus missing rate to minimise missing data among individuals. This resulted in eliminating intraspecific diversity in *D. exilis* and *D. longiflora*, i.e. the SNPs present in these species but not in *D. iburua* and *D. ternata*. Supplementary Table 2 indicates that using a less stringent threshold on the locus missing rate (20%) increased the estimates of genetic diversity π for *D. exilis* and *D. longiflora*, thus reducing the difference in genetic diversity between the two pairs of species."

Filtering strong on missing SNPs when one species does not map well, will lead to even stronger bias. Particularly as the number of samples in the wild species is very low. For instance with your 5% filter across all samples an SNP with an allele frequency of 50% (which is high) in the wild population, but absence in the reference population will be removed and not used to calculate diversity.

Filtering for diversity measures should be considered within species. In this light it is actually complicated to understand how samples with N=11 can have such high diversity. It might well explain the low differentiation between species though. It's probably worth looking at allelic richness and the number of private alleles per species.

The k-mer approach might be helpful, and it might make sense to exchange the tables in the main text.

Our response:

Thank you for your feedback and for clearly explaining your concerns about our genetic diversity estimates. We agree with you that filtering for the locus missing rate within species will lead to more robust and relevant estimates of genetic diversity. Consequently, we decided to perform more appropriate computations of diversity statistics for each species and to integrate the k-mer approach into the main text as suggested.

For the calculation of diversity statistics with the SNP dataset, whereas in the previous version we used three different thresholds (5%, 10%, 20%) for the locus missing rate considering all samples, we have now filtered with a unique threshold (20%) for each species separately. Moreover, we decided to compute diversity statistics across the genomes using sliding windows of size 50kb and step sizes of 10kb. This allows us to estimate the variance of each diversity statistics and compare the values statistically.

For the k-mer approach, we used the Jaccard index as a measure of diversity within populations. The index is based on the proportion of shared/non-shared k-mers within the population, and the results were obtained for 100 replicates and random samples of 10,000 k-mers.

We thus modified accordingly the “Population diversity statistics” paragraph of the methods section:

*[lines 492-505]: “We computed the number of segregating sites (S), the nucleotide diversity (π), Watterson’s Θ (Θ_w) and Tajima’s D per species with `egglib v.3.2.1`⁶², which is a Python module that enables fast computation of population genomic diversity statistics. Statistics were computed **across the genomes using sliding windows of size 50kb and step sizes of 10kb. A threshold of 20% for the per-locus missing rate was applied for each species separately.** ~~with the three SNP datasets filtered at different thresholds for the per-locus missing rate locus missingness (5%, 10% and 20%).~~ LD decay (r^2) was computed per species with `PopLDdecay v.3.42`⁶³ within a 500 kb distance. We then used the Perl script `Plot_MultiPop.pl` and `ggplot2 v.3.4.2`⁶⁴ R package to plot the figures.*

The Jaccard index (JI) was calculated with the k-mer dataset as a measure of genetic diversity within species without mapping bias. We estimated a dissimilarity index based on the proportion of shared and non-shared k-mers from the 1,000,000 presence/absence table, which was split into 100 tables of random samples of 10,000 k-mers from which the JI values were obtained.”

For the results, the following figure has been included as Figure 2 in the main text and supplementary tables displaying mean values of the different diversity statistics with statistical tests have also been included. We have also corrected a small typo in the numbers of individuals reported in the tables and used for the calculations. For *D. exilis*, the correct number is 199 instead of 200. For *D. iburua*, the number is 21 instead of 22. These errors concerned only the presentation of the figures and not the calculations themselves.

Fig. 2 Patterns of genetic diversity in fonio millets and wild relatives, with and without consideration of the *D. exilis* reference genome for calculations. a. Boxplot distribution of nucleotide diversity (π) computed using sliding windows of size 50kb and step sizes of 10kb. b. Boxplot distribution of the Jaccard dissimilarity index within species and generated from the 1,000,000 k-mers presence/absence table. Statistical comparisons between each pair of population are presented in Supplementary Table 3.

In the main text, the new sentences are:

[lines 136-152]: “*The nucleotide diversity (π) of cultivated *D. exilis* and *D. iburua* was lower ($P < 2.2 \times 10^{-16}$; two-tailed Welch *t*-test, Supplementary Table 3) than that of their wild relatives *D. longiflora* and *D. ternata* (Fig. 2a, Supplementary Table 2). Moreover, black fonio displayed higher ($P < 2.2 \times 10^{-16}$; two-tailed Welch *t*-test, Supplementary Table 3) genetic diversity than white fonio and a lower reduction in nucleotide diversity compared to its wild relative *D. ternata* (Fig. 2a, Supplementary Table 2). These patterns were also obtained with the Jaccard dissimilarity index that can be viewed as a genome reference-free measure of the population genetic diversity computed with *k*-mers (Fig. 2b). However, the mean values between *D. iburua* and *D. ternata* were not significantly different ($P = 0.15$; two-tailed Welch *t*-test, Supplementary Table 3). The Watterson’s Θ values were higher than π for both species*

of the *D. exilis*/*D. longiflora* pair, while being lower for the *D. iburua*/*D. ternata* pair. A negative Tajima's *D* value was obtained for *D. exilis*/*D. longiflora*, while being positive for *D. iburua*/*D. ternata* (Supplementary Table 2). We identified linkage disequilibrium (LD) decays according to the kb distance for *D. exilis* and *D. longiflora*, while LD very quickly decayed to its minimum value for *D. iburua* and *D. ternata* (Supplementary Fig. 3)."

Supplementary Table 2. Genetic diversity estimates computed for each species with SNP datasets, applying a filter of 20% for the locus missing rate, for each species separately. Statistics were computed using sliding windows of size 50kb and step sizes of 10kb. Mean values with standard deviation are shown.

	D. exilis	D. longiflora	D. iburua	D. ternata
#ind	199	14	21	11
#sites	15,257,883	9,693,764	2,590,230	1,546,299
π	0.019 (SD = 0.023)	0.155 (SD=0.036)	0.107 (SD=0.040)	0.155 (SD=0.49)
Θ	0.033 (SD=0.021)	0.164 (SD=0.034)	0.083 (SD=0.026)	0.146 (SD=0.040)
Tajima's D	-1.44 (SD = 1.11)	-0.21 (SD=0.42)	0.98 (SD=0.77)	0.26 (SD=0.52)
S	3,151,567 (20.66%)	5,946,094 (61.34%)	951,947 (36.75%)	836,109 (54.07%)

#ind = Number of considered individuals; #sites = Total number of sites across the genome with a locus missing rate < 20% for the species considered; π = Nucleotide diversity; Θ = Watterson's estimator of nucleotide diversity; S = Total number (and proportion) of polymorphic sites across the genome with a locus missing rate < 20%.

Supplementary Table 3. Statistical comparisons of genetic diversity statistics for each pair of populations. Two-tailed Welch t-test were performed.

Diversity statistic	Species pair	t-statistic	Degrees of freedom	p-value
π	D. exilis / D. longiflora	-802.11	105811	< 2.2e-16
	D. exilis / D. iburua	-484.13	99379	< 2.2e-16
	D. exilis / D. ternata	-630.54	87051	< 2.2e-16
	D. longiflora / D. iburua	224.33	124424	< 2.2e-16
	D. longiflora / D. ternata	-2.0232	114096	0.04305
	D. iburua / D. ternata	-191.78	119035	< 2.2e-16
Θ	D. exilis / D. longiflora	-826.25	106982	< 2.2e-16
	D. exilis / D. iburua	-369.8	121511	< 2.2e-16
	D. exilis / D. ternata	-622.12	95239	< 2.2e-16
	D. longiflora / D. iburua	483.57	118052	< 2.2e-16
	D. longiflora / D. ternata	90.247	121295	< 2.2e-16
	D. iburua / D. ternata	-331.74	106559	< 2.2e-16
Jaccard index (k-mers)	D. exilis / D. longiflora	-30.484	183.84	< 2.2e-16
	D. exilis / D. iburua	-35.162	182.72	< 2.2e-16
	D. exilis / D. ternata	-30.188	154.61	< 2.2e-16
	D. longiflora / D. iburua	-13.511	155.26	< 2.2e-16
	D. longiflora / D. ternata	-12.341	132.41	< 2.2e-16
	D. iburua / D. ternata	-1.4552	183.22	0.1473

The old sentences were:

“The mean nucleotide diversity (π) of cultivated *D. exilis* was approximately 100-fold lower than that of its wild relative *D. longiflora* (Table 1). On the other hand, *D. iburua* and *D. ternata* shared higher and similar nucleotide diversity values (Table 1). The Watterson’s Θ values were higher than π for both species of the *D. exilis*/*D. longiflora* pair, while being lower for the *D. iburua*/*D. ternata* pair. A negative Tajima’s *D* value was obtained for *D. exilis*/*D. longiflora*, while being positive for *D. iburua*/*D. ternata* (Table 1). We identified linkage disequilibrium (LD) decays according to the kb distance for *D. exilis* and *D. longiflora*, while LD very quickly decayed to its minimum value for *D. iburua* and *D. ternata* (Supplementary Fig. 3). With less stringent thresholds for the locus missing rate (10% and 20% instead of 5%), the genetic

diversity estimates were still higher for D. iburua and D. ternata than for D. exilis and D. longiflora. The Tajima's D values were still negative for the D. exilis/D. longiflora pair and positive for the D. iburua/D. ternata pair (Supplementary Table 2)."

In the discussion of the previous version of the article, we explained these genetic differences between the two species pairs with the use of the *D. exilis* reference genome and potential differences in reproduction systems:

"The genetic diversity of black fonio (D. iburua) and its wild relative (D. ternata) was much higher than for white fonio (D. exilis) and D. longiflora. This was probably due to the use of the D. exilis reference genome which differs markedly from the D. iburua and D. ternata genomes, or it could reflect differences in reproduction systems among the two species pairs."

In fact, we believe that the use of the *D. exilis* reference genome did not inflate estimates of genetic diversity in *D. iburua* and *D. ternata*, but rather underestimated it. We thus modified accordingly the paragraph on genetic diversity and we moved it up in the text, just after the first sentences on population structure.

New sentences are:

*[lines 268-284]: "The genetic diversity of black fonio (D. iburua) and its wild relative (D. ternata) was higher than for white fonio (D. exilis) and D. longiflora. **This result was even more pronounced with the Jaccard index computed with k-mers.** The use of the D. exilis reference genome which differs markedly from the D. iburua and D. ternata genomes **probably underestimated their genetic diversity, since low mapping rates may lead to the loss of genetic variation detected during SNP calling. These differences in genetic diversity could reflect differences in reproduction systems among the two species pairs. In fact, the self-fertilization rate of D. exilis is around 99%³³. Outbreeding might actually be more pronounced for black fonio and its wild relative, thus increasing their genetic diversity. More severe bottlenecks associated with the domestication and diffusion history of white fonio could also explain its lower diversity compared to black fonio. Given our results, we recommend using k-mers as a validation criterion in further genomic studies on indigenous crops (which often lack a reference genome), but also as a means to study their genetic diversity. Such approaches are receiving increasing attention in the of population genomics field when no reference genome is available³⁴.**"*

The old sentences were:

*“The genetic diversity of black fonio (*D. iburua*) and its wild relative (*D. ternata*) was much higher than for white fonio (*D. exilis*) and *D. longiflora*. This was probably due to the use of the *D. exilis* reference genome which differs markedly from the *D. iburua* and *D. ternata* genomes, or it could reflect differences in reproduction systems among the two species pairs. In fact, the self-fertilization rate of *D. exilis* is around 99%³⁵. Outbreeding might actually be more pronounced for black fonio and its wild relative, thus increasing their genetic diversity. The genetic diversity of populations and species could potentially be studied using indices calculated with kmers, such as the Jaccard dissimilarity index³⁶, which allows calculation of the proportion of shared kmers within a population. Such approaches are receiving increasing attention in the of population genomics field when no reference genome is available³⁷.”*

Finally, we looked at allelic richness (Figure R1) and the number of private alleles per species (Table R1). We identified private alleles based on SNPs that are presents at least once in each species. We think that this information is redundant with the diversity statistics considered in the manuscript and is not as relevant to include as the figure presenting the diversity calculated with k-mers.

Figure R1. Allelic Richness per species averaged across all considered sites that have a missing rate < 20%. For each site, the allelic richness was calculated with egglib according to the following formula: (number of alleles – 1) / (number of samples – 1).

Table R1. Summary of private alleles for the four species.

Species	#individuals	#sites	Private SNPs	% Private
Digitaria exilis	199	7,644,765	440,949	5.77
Digitaria longiflora	14	7,644,765	2,686,232	35.14
Digitaria iburua	21	7,644,765	437,921	5.73
Digitaria ternata	11	7,644,765	1,120,693	14.66